# Layer-By-Layer Self-Assembled Dip Coating for Antifouling Functionalized Finishing of Cotton Textile

**DOI:** 10.3390/polym14132540

**Published:** 2022-06-22

**Authors:** Sana Javaid, Azhar Mahmood, Habib Nasir, Mudassir Iqbal, Naveed Ahmed, Nasir M. Ahmad

**Affiliations:** 1School of Natural Science (SNS), National University of Science and Technology (NUST), Islamabad 44000, Pakistan; sana.javaid@sns.nust.edu.pk (S.J.); dr.azhar@sns.nust.edu.pk (A.M.); habibnasir@sns.nust.edu.pk (H.N.); mudassir.iqbal@sns.nust.edu.pk (M.I.); 2Department of Chemistry, University of Wah, Wah Cantt 47040, Pakistan; 3Department of Pharmacy, Quaid-i-Azam University, Islamabad 44000, Pakistan; natanoli@qau.edu.pk; 4Polymer Research Lab, School of Chemical and Materials Engineering (SCME), National University of Sciences and Technology (NUST), Islamabad 44000, Pakistan

**Keywords:** antifouling polymeric formulation (APF), finishing, nanoencapsulation, cotton textile, layer-by-layer technique

## Abstract

The fouling of surfaces such as textiles is a major health challenge, and there is a continuous effort to develop materials and processes to overcome it. In consideration of this, this study regards the development of antifouling functional nanoencapsulated finishing for the cotton textile fabric by employing a layer-by-layer dip coating technique. Antifouling textile finishing was formulated by inducing the nanoencapsulation of the antifouling functional group inside the hydrophobic polymeric shell. Cotton fabric was taken as a substrate to incorporate antibacterial functionality by alternatively fabricating multilayers of antifouling polymeric formulation (APF) and polyelectrolyte solution. The surface morphology of nanoencapsulated finished textile fabric was characterized through scanning electron microscopy to confirm the uniform distribution of nanoparticles on the cotton textile fabric. Optical profilometry and atomic force microscopy studies indicated increased surface roughness in the coated textile substrate as compared to the uncoated textile. The surface thickness of the fabricated textile increased with the number of deposited bilayers on the textile substrate. Surface hydrophobicity increased with number of coating bilayers with θ values of x for single layer, up to y for 20 bilayers. The antibacterial activity of the uncoated and layer-by-layer coated finished textile was also evaluated. It was significant and exhibited a significant zone of inhibition against microbial strains Gram-positive *S. aureus* and Gram-negative *E. coli.* The bilayer coating exhibited water repellency, hydrophobicity, and antibacterial activity. Thus, the fabricated textile could be highly useful for many industrial and biomedical applications.

## 1. Introduction

Textile substrates are susceptible to the growth of bacteria, fungi, and various other microbes [1]. The proliferation rate of bacteria and other microbes is aggravated under suitable conditions of optimal temperature, pH, humidity, and moisture. The deterioration and biofouling of textile material in industrial and medical or health sectors pose serious threats. The spread of pathogenic microbes through contaminated textiles is a major source of nosocomial infections [2]. Gram-negative and Gram-positive bacteria species such as *Escherichia coli* and *Staphylococcus aureus* are among those bacterial strains responsible for hospital-acquired infections [3,4,5]. In consideration of such concerns, the global demand for antifouling textiles has tremendously increased due to consumer awareness regarding hygiene, health, and safety [6]. The annual production of medical textiles is predicted to significantly increase to up to 3500 metric tons by 2025 [7].

Textile functionalization through antifouling agents has allowed for many technological applications in biomedical fields, such as bandages, wound dressing, artificial sutures, body-implanted and nonimplanted devices, and tissue scaffolding [8,9]. Nanotechnology plays a significant role in the functionalization and modification of textiles by incorporating immobilized nanoparticles on the textile surface by encapsulating active agents [10]. Advancements in nanoencapsulation for functional finishes impart dyeing, anti-UV, flame-retardant, insect- and moth-repellent, antibacterial, self-healing, and aroma-producing properties to textiles [11,12,13,14].

Polymeric natural and synthetic nanomaterial is an efficient carrier of active agents to specific sites owing to its biodegradability, biocompatibility, and nontoxicity [15]. Polycaprolactone fulfills the requirements of bioapplications with suitable features for encapsulating antifouling functionalities [16,17]. Because of the above, the development and modification of biodegradable PCL nanoparticles for the slow release of an active agent through nanoencapsulation are very important, particularly in skin applications [18,19,20]. Different approaches for achieving nanoencapsulation by using biodegradable polymeric shells and bioactive core material include emulsification through coacervation, diffusion, or evaporation, and interfacial and in situ polymerization [21,22,23,24]. Both physical and chemical methods for nanoencapsulation have been studied. Natural cotton textiles incorporating nanocapsules for imparting antibacterial properties require attention for hygienic wound dressing [25]. Eugenol-loaded human serum albumin/silk fibroin (HSA/SF) nanocapsules with antibacterial properties were synthesized for the functionalization of cotton/polyethylene terephthalate blends. Functionalization was achieved for cotton wound dressing with significant antibacterial properties against *Escherichia coli* and *Staphylococcus aureus*. The in vitro release of pH-sensitive antimicrobial agents from nanocapsules reached the textile surface to inhibit bacteria [26]. Antibacterial formulations are generally coated on textiles with various techniques such as pad–dry–cure (dipping method) or padding, spray coating, and the sol-gel technique [27]. Padding is the most conventional treatment for functionalized, but it is difficult to manipulate the effect of shearing forces acting on textile surfaces of due to the large number of process parameters. Among various coating techniques, the layer-by-layer (LBL) deposition of polyelectrolytes for the antifouling functional finishing of cotton is the simplest, easiest, and environmentally friendly approach [28]. Regarding this, chitosan/pentasodium tripolyphosphate as antibacterial functionality was coated on ionized cotton using the LBL deposition of oppositely charged polyelectrolytes. The subsequent deposition of multiple bilayers consequently increased the antibacterial efficacy of cotton against *E. coli* and *S. aureus* [29].

Considering the significance of antibacterial functional textiles for biomedical and industrial applications, we employed nanoencapsulation for antifouling finishing. Nanocapsules were prepared with biodegradable polycaprolactone (PCL)-encapsulated cefotaxime as antifouling moiety, and they were immobilized on a cotton textile. PCL is semicrystalline and biocompatible, achieving significant performance in the slow release of active molecules on fabric surfaces [30]. Cefotaxime, a third-generation semisynthetic cephalosporine, was chosen as an antifouling functional moiety having a broad range of antibacterial activity, particularly against *E. coli* and *S. aureus*. The efficacy of cefotaxime in bactericidal action by inhibiting the synthesis of peptidoglycan is associated with the high stability of its beta-lactamase. The toxicity of cefotaxime was significantly reduced by encapsulation, and caused the slow and sustained release that is particularly suitable for fabricating functional textiles. The molecular structure of cefotaxime is presented in Figure 1.

Antifouling polymeric formulations (APFs) were previously developed through nanoprecipitation and employed to fabricate antifouling textiles. Thus, the layer-by-layer technique was used to develop self-assembled multilayers through the dip coating of an antifouling formulation to fabricate cotton textiles for the in vitro and gradual release of antifouling activity. A cotton substrate was selected as the model textile fabric for antifouling due to its hydrophilicity and rapid deterioration in moist environments to allow for microbial degradation. The layer-by-layer coating of a polymeric formulation on cotton textile fabric induced significant hydrophobicity. The fabricated textile was analyzed through various characterization techniques. Surface wettability was analyzed by measuring the contact angle (θ), and surface roughness, thickness, and morphology were studied through optical profilometry, atomic force microscopy (AFM), and scanning electron microscopy (SEM). In vitro antibacterial tests were conducted to assess antifouling properties. The efficacy of antifouling activity of the developed cotton textile was dependent on the concentration of APF, deposited via self-assembled bilayers. The fabricated textile showed antibacterial activity against Gram-positive *Staphylococcus aureus* and Gram-negative *Escherichia coli* with a significant reduction in bacterial colony numbers. The novel nanoencapsulation finishing could be used in developing woven and nonwoven textiles with antibacterial functionalities. It could be utilized to fabricate various types of industrial smart textiles, including cotton wound dressing for healing skin irritations and skin allergies.

## 2. Materials and Methods

Polycaprolactone (PCL) (Mw-1400 g/mol), polyvinyl alcohol (PVA) (Mw 3100 g/mol), poly (diallyl dimethyl ammonium chloride) (PDAC) (average Mw 200,000–350,000, 20 wt% solution), and dichloromethane (DCM) (Mw-84.93) with 99.9% purity were purchased from Sigma Aldrich, Germany. Deionized ultrapure water with total dissolved solids (TDS~0.00) was used for solution preparation and washing. Cefotaxime (Mw 455.47 g/mol in powder form) as an antifouling functional moiety was obtained from Nectar Life sciences, India. Mercerized bleached (100%) whitish pure cotton with areal density of 1.31 g/m^3^ was obtained from the National Textile University (NTU), Faisalabad, Pakistan. Sodium chloride (NaCl), purchased from Sigma Aldrich (Germany), was used as a buffer saline solution. Mueller-Hinton II Agar (MHA) (Biolab, Budapest, Hungry) was utilized as the nutrient broth for the bacterial culture. In vitro antibacterial assay was performed against clinical bacterial strains of Gram-positive *Staphylococcus aureus* (ATCC 6538) and Gram-negative *Escherichia coli* (ATCC 8739).

### 2.1. Synthesis of Antifouling Polymeric Formulation

The antifouling polymeric formulation (APF) was synthesized and characterized of as described in previous work [31]. The nanoencapsulation of antifouling functional cefotaxime was conducted by using polycaprolactone as the shell material through nanoprecipitation. The organic phase containing polycaprolactone (PCL) was dissolved in dichloromethane (DCM), while the aqueous phase containing polyvinyl alcohol (PVA) was used as a stabilizing agent. To optimize the blank formulation, an organic phase was slowly injected into an aqueous phase. After achieving a stable formulation by varying the process parameters, cefotaxime was dissolved in the organic phase and slowly injected into an aqueous phase. The subsequent addition of the organic phase containing cefotaxime into an aqueous phase formed the antifouling polymeric nanoformulation.

### 2.2. Characterization of Antifouling Polymeric Formulation

The prepared antifouling polymeric nanoformulation was analyzed with different techniques. Surface morphology was studied with scanning electron microscopy (SEM) (model JEOL JSM 6490 LA, Tokyo, Japan); applied voltage was 10 KV. The encapsulation of antifouling functionality in the polymeric nanoparticles was evidenced with Fourier transform infrared (FTIR) using an FTIR spectrophotometer (Modal Shimadzu 8400, Markham, ON, Canada). Dynamic light scattering (DLS) (Nano ZS, Malvern Instruments, Worcestershire, UK) was used to observe the average size, surface charge, and distribution of nanoparticles.

### 2.3. Fabrication of Antifouling Polymeric Coating on Textile

Pretreated and mercerized cotton textile fabric was received as plain whitish woven with an areal density of 1.31 g/m^3^. The antifouling polymeric coating on the cotton textile was fabricated through a layer-by-layer technique. Cotton textile fabric was alternatively dipped in a polycation solution of 10^−3^ M poly(diallyl dimethyl ammonium chloride) (PDAC) solution and 100 mL of diluted antifouling polymeric formulation (APF) comprising 25 mg of PCL in a series of steps. Antibacterial tests were conducted against clinical isolates of Gram-positive *Staphylococcus aureus* (ATCC 6538) and Gram-negative *Escherichia coli* (ATCC 8739).

The pretreated cotton textile was cut into swatches of 75 by 25 mm. Negatively charged cotton swatches were first dipped in PDAC (a polycation solution maintained at neutral pH) for 10 min, resulting in positive charge deposition (single ion layer) followed by subsequent washing with deionized water twice for 5 min in two separate beakers to achieve the uniform deposition of charges, and to remove unbound or loosely attached molecules from the textile surface. For the opposite charge of another single ion layer, cotton swatches were dipped in an antifouling polymeric formulation (APF maintained at neutral pH) for 10 min, followed by washing with deionized water twice for 5 min in two separate beakers. Thus, a bilayer (1bL) of oppositely charged ions was formed due to electrostatic interaction on the cotton textile between the positively charged polyelectrolyte and negatively charged antifouling polymeric formulation (APF). The whole process of fabricating a single bilayer was accomplished in 40 min. The same procedure was repeated to cyclically fabricate 5, 10, 15, and 20 bilayers without the textile drying in each step. Fabricated textiles with top layers of negatively charged APF were considered for the slow release of antifouling functionality and characterized for an in vitro antibacterial test. The schematic representation of the layer-by-layer coating of the cotton textile is given in Figure 2.

### 2.4. Characterization of Fabricated Cotton Textile Fabric

The layer-by-layer coated textile fabric was analyzed by different techniques. The antifouling finished cotton textile fabric was analyzed for surface morphology, thickness, and roughness. The surface morphology of the coated and uncoated textile samples was observed through SEM (Modal JSM 6490LA, JEOL, Tokyo, Japan). Textile swatches of equal dimensions were fixed on the glass slide, followed by sputter-coating with gold for analysis. Optical profilometry was used for measuring the difference between the average roughness and thickness of the uncoated and coated textiles in micrometers. We fixed 20 × 5 mm textile swatches at the center of the glass slide with double tape, and analyzed them through a 2D noncontact profilometer (model NANOVEA PS-50, USA). The subsequent roughness due to the layer-by-layer deposition of the nanoformulation inducing nanoencapsulated finished cotton textile fabric was further confirmed through scanning probe microscopy (Modal JSPM-5200, JEOL, Japan). Then, 20 × 5 mm textile swatches were cut and fixed on the glass slides with the double tape, and were analyzed in tapping mode. Uncoated and coated textile samples were topographically analyzed through 3D plots for detailed visualization. The surface wettability and hydrophobicity of the fabricated cotton textile were analyzed by measuring the contact angle (θ). Textile swatches (20 × 5 mm dimension) were clipped at the center of the glass slide and placed under a drop-shape analyzer (Model Kruss Gmbh 2014–2020, Hamburg, Germany).

### 2.5. In Vitro Antibacterial Bioassay

Standard qualitative textile testing with AATCC TM 100 with slight modification was performed to assess antibacterial activity over the course of 24 h [32]. Clinical isolates of *Staphylococcus aureus* (ATCC 6538) and *Escherichia coli* (ATCC 8739) were utilized to check the antibacterial activity of the fabricated cotton textile by agar disk diffusion or Kirby–Bauer assay. For this purpose, a Muller Hinton Agar (MHA) solution was prepared as commercially recommended and autoclaved overnight. The freshly prepared nutrient broth was streaked with microbial cells and incubated overnight at 37 °C. Then, a 10 mL saline solution was poured into a 50 mL sterilized falcon tube and inoculated with pathogenic colonies, followed by vortexed mixing for constant distribution. Afterwards, 0.5 McFarland was utilized to set the optical density and turbidity [33]. MHA (25 mL) solution was poured in sterilized Petri dishes and it was allowed to solidify the agar; then, it was placed in an incubator at 37 °C overnight. Solid agar plates were streaked with sterile cotton swabs by adding the bacterial culture (about 50 microliters). Streaking was conducted by rotating the Petri plate about 60 degrees thrice for uniform inoculation. Uncoated and coated (1, 5, 10, 15, and 20 bl) textiles of 40 × 10 mm dimensions were placed onto solid agar plates by gently pressing the textile at the center of each Petri dish. Plates were placed in an incubator at 37 °C overnight, and antibacterial activity was observed after 24 h. The zone of inhibition around the textile sample was measured as mean standard deviation.

## 3. Results and Discussion

### 3.1. Characterization of Antifouling Polymeric Formulation (AFP)

Cefotaxime was nanoencapsulated as a core antifouling functionality inside the polymeric shell for the slow release of in vitro antibacterial activity as described previously [31]. Polycaprolactone nanoparticles were characterized for average size, surface charge, and morphology. The average size and zeta potential of the drug-loaded polymeric nanoparticles were 216 nm and −11.1 mV, respectively, as shown in Figure 3. Uniformity in particle size distribution was observed at a PDI of 0.4 (less than 0.5 shows uniformity in distribution). The negatively charged zeta potential of PCL nanoparticles was attributed to the hydrolysis of the carboxyl and hydroxyl groups present at the surface.

### 3.2. Characterization of Antifouling Fabricated Cotton Textile

#### 3.2.1. Surface Morphology Studies

The surface morphology of layer-by-layer-deposited thin films of antifouling polymeric (APF) coating was studied by observing the SEM micrographs. Cross-sectional views of SEM micrographs showed the surface morphology of the uncoated and coated cotton textiles, as given in Figure 4. The smooth and uniform surface of the uncoated mercerized cotton textile fabric is shown in Figure 4a, while a significant increase in roughness in the morphology of the coated textile appeared after coating 1 bilayer up to 20 bilayers as shown in Figure 4b–f. The multilayer coating of polyelectrolytes over the cotton textile surface produced an agglomeration [34]. The density of agglomerates increased with the sequential adsorption of the number of bilayers. SEM micrographs of the coated textile fabric showed the uniform and homogeneous deposition of nanoparticles over the surface. There were fewer agglomerates with the coating of 1 and 5 bilayers, as shown in Figure 4b,c, as compared with the coated textile fabric with 10, 15, and 20 bilayers, as shown in Figure 4d–f, with a subsequent increase in the population of agglomerates. The less and dense agglomeration of charged particles over the textile surface depends upon various factors, including the nature and concentration of the polyelectrolyte solution (cationic or anionic polyelectrolyte), immersion time in deionized water, polyelectrolytes, and nanoformulations, besides the cyclical drying process in each step [35]. The roughened surface of the fabricate textile fabric after subsequent bilayer coating potentially created hydrophobicity and antifouling activity [36].

The surface roughness of the functionalized cotton textile from the layer-by-layer dip coating of polyelectrolyte solution and APF formulation was observed through optical profilometry. Mercerized/bleached cotton was smooth, shiny, and whitish before coating. There was significantly increased average roughness (Ra) in the deposited thin film over cotton textile [37]. The average roughness (Ra) of the uncoated (0 BL) and coated textiles (1, 5, 10, 15, and 20 BL) was calculated in µm with a 2D noncontact profilometer. A marked difference in the roughness profile of the uncoated and coated textiles was attributed to the incremental deposition of antifouling polymeric formulation (APF) and poly(diallyl dimethyl ammonium chloride) (PDAC) solution. The comparatively smooth surface of uncoated textile and increasing trend in average roughness of the coated textiles with a deposited number of bilayers are shown in Figure 5. This was also evidenced by the scanning electron micrographs of the uncoated and coated textiles [38].

The surface topography of the uncoated and coated textile samples was further studied by using atomic force microscopy (AFM). The topography of the uncoated and coated textiles was better visualized by scanning a small area and taking 3D plots, as shown in Figure 6. Tapping mode AFM was used for the topographical analysis of the textile samples by scanning an area in µm. The marked difference in the roughness profile of the uncoated and coated textiles was quantitively analyzed in terms of root mean square roughness (Rq) by plotting a graph between Rq values and the number of deposited bilayers on the cotton textile fabrics, as shown in Figure 7. Rq values were evaluated in terms of height or thickness as measured from the mean level of surface [39]. The value of Rq increased with the increase in the number of deposited bilayers, confirming that the Rq value varies with the composition and density of the sample [40]. Topographical AFM 3D plots showed characteristic bright and dark portions, with brighter groves indicating the deposition of nanoparticles. As the number of deposited bilayers increased, the bright portion of the peaks and valleys groves became more significant in the case of the coated textiles with 5, 10, 15, and 20 bilayers. These bright groves and thickness in peak and valleys resulted due to the electrostatic interaction between the tip and surface of the textile. The denser the textile surface with the increased number of bilayers was, the more significant the portion of bright groves with characteristic roughness that appeared was. The resulting variation depended upon the alternative deposition of the antifouling polymeric formulation (APF) and PDAC solution in the modified textile samples [41]. The mercerized cotton was smooth and shiny, while the nanoencapsulated finished cotton textile induced the aggregation of nanoparticles with the addition of some bilayers on the textile surface, thus enhancing micro- and nanoscale roughness [38,42]. The acquired roughness (Rq) of the nanoencapsulated finished cotton textile fabric was evidenced by contact angle measurement for water repellency and the desirable antifouling properties.

#### 3.2.2. Surface Wettability Analyses

The surface wettability of thenanoencapsulated finished cotton textile fabric was studied by measuring the contact angle (θ) as shown in Figure 8. The hydrophilicity and hydrophobicity of the fabricated textile was quantitatively analyzed by observing the values of the contact angle (θ) and surface energy (γ_s_). The surface energy of the finished textile substrates was calculated from the contact angle measurement by considering the interaction at the interface of solid and liquid surfaces [43,44]. Variation in contact angle (θ) and surface energy (γ_s_) with the number of deposited bilayers in the nanoencapsulated finished textile is shown in Figure 9a,b respectively. The uncoated cotton textile with a low contact angle (θ) had a mercerized fine cotton texture and the presence of hydroxyl group that lead to maximal hydrophilicity and high surface energy (γ_s_). Moreover, the increased value of surface energy was associated with pronounced wettability and maximal surface coverage, and this could lead to a poor antifouling effect [36]. The water contact angle (θ) of the 1 bilayer coated textile was greater than that of the uncoated textile substrate due to the hydrophobic nature of PCL. The increased value of the contact angle (θ) resulted from the deposition of 5 bilayers, and was associated with the roughness and uneven protrusions on the textile substrate [45]. Further coating with 10, 15 and 20 bilayers showed an increased value of θ with differences among them as compared to the 1- and 5-bilayer-coated textile due to the interpenetration of the next bilayer with the previously adsorbed bilayer [46]. Thus, a greater number of bilayers coating led to nonuniform deposition and inhomogeneous surface of the textiles. Therefore, water repellency and comparative hydrophobicity were a result of the deposited bilayers in the nanoencapsulated finished cotton textile fabric.

The surface energy of the fabricated cotton textile was evaluated through the results of contact angle (θ) [47]. Thus, Young’s model summarizes the theoretical description in Young’s equation (γ_s_ = γ_s_ -_l_ + γ_l_ cosθ), a best fit for smooth surfaces before coating applications. γ_s_ stands for surface energy of solids, γ_s_ -_l_ is the surface energy of the solid–liquid interface, γ_l_ is the surface energy of liquids, and θ is the contact angle of a liquid droplet. The developed layer-by-layer dip coating of antifouling polymeric formulation (APF) induces significant water repellency associated with the hydrophobicity of PCL [48].

#### 3.2.3. Antibacterial Assay of Textile Fabric

The antibacterial activity of the uncoated and nanoencapsulated finished cotton textiles was established against two bacterial strains, Gram-positive *S. aureus* and Gram-negative *E. coli*, by using a qualitative agar disk diffusion assay [49]. The antibacterial test results of the uncoated and coated textile fabrics against *E. coli* and *S. aureus* are shown in Figure 10 and Figure 11 respectively. The uncoated textile was taken as a control against both strains to compare the antibacterial efficacy of the nanoencapsulated finished cotton textile fabricated with the layer-by-layer coating of the antifouling polymeric formulation (APF). The antibacterial efficacy of fabricated finished textile was analyzed by placing the textile swatches (40 × 10 mm) in intimate contact on agar plates prestreaked with bacterial inoculum [50]. A clear zone of inhibition was observed around the modified textiles coated with 1, 5, 10, 15, and 20 bilayers. The zone of inhibition against both strains progressively increased around the textile swatches fabricated with a progressive coating of bilayers. These results were attributed to the significant release of antifouling functional drugs from the polymeric nanoencapsulated finished textile in a concentration-dependent manner [31,51]. The release of antifouling drug cefotaxime from APF was responsible for the antifouling activity of the fabricated textiles. The antibacterial activity of the fabricated textiles was further compared against both strains with a greater zone of inhibition against *E. coli* as compared to *S. aureus,* as shown in Figure 12. Thin-walled Gram-negative *E. coli* was more susceptible to bactericidal action than Gram-positive *S. aureus* was. Moreover, there was electrostatic repulsion between Gram-negative *E. coli* and the negatively charged layer of the antifouling polymeric formulation (APF) coated on the textile [52]. The antifouling activity of the nanoencapsulated finished textile fabric was attributed to the slow release of cefotaxime encapsulated inside the PCL shell. As the number of deposited bilayers increased, antifouling activity increased due to the increased release of cefotaxime form PCL nanoparticles. Thus, the antifouling polymeric formulation (APF) encapsulated nanoparticles of cefotaxime, as a notable antibacterial moiety has significant bactericidal action attributed to the high stability of beta-lactamase enzymes, thus inhibiting the synthesis of bacterial cell walls. It has significant bactericidal action against Gram-negative bacterial strains, while it is associated with adequate action towards Gram-positive bacteria [53].

## 4. Conclusions

The growing demand for health and hygiene regarding textile products has led to industrial concerns about antifouling textiles. Thus, the present work illustrates a novel approach to fabricating nanoencapsulated finished cotton textile fabrics by incorporating the antifouling functional group finishing of cotton textile achieved by employing a layer-by-layer technique through the alternate dip coating of oppositely charged polyelectrolyte solution and antifouling polymeric formulation (APF). SEM and AFM images confirmed the significant deposition of polymeric nanoparticles with the increasing number of coating bilayers. The optical profilometry analysis of the functionalized textiles showed that the multilayer deposition of thin films resulting in the subsequent coating of 1 to 20 bilayers increased surface roughness from 16.4 to 69.4 µm. Further, the surface wettability of the functionalized textiles was evaluated by measuring the contact angle (θ), which increased from 23° to 48.5°, and surface energy decreased from 145 to 18 dynes/cm, which showed the decreased wettability and increased surface hydrophobicity associated with the deposited bilayers. The nanoencapsulated finished textiles were also analyzed through an in vitro antibacterial assay against Gram-negative *E. coli* and Gram-positive *S. aureus,* with a significant reduction in the bacterial cell colonies of both strains. Moreover, the bactericidal efficacy of the fabricated textiles was evaluated in a concentration-dependent manner, and showed an exponential zone of inhibition with the number of coating bilayers on the textiles. The zone of inhibition against both strains increased with the number of deposited bilayers due to the increased release of cefotaxime from the polymeric nanoparticles. The resulting antifouling functional finishing, achieved through the nanoencapsulation of the antifouling functional group, could provide a substantial route in fulfilling the growing global needs for antifouling textiles. The nanoencapsulated finished antifouling cotton textile fabric could be used in wound dressing, for skin infections, and in various other industrial applications.

## Figures and Tables

**Figure 1 polymers-14-02540-f001:**
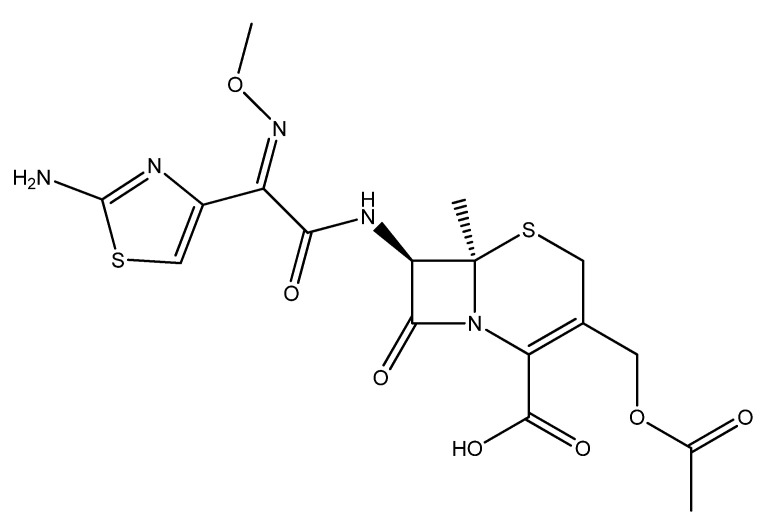
Molecular structure of cefotaxime.

**Figure 2 polymers-14-02540-f002:**
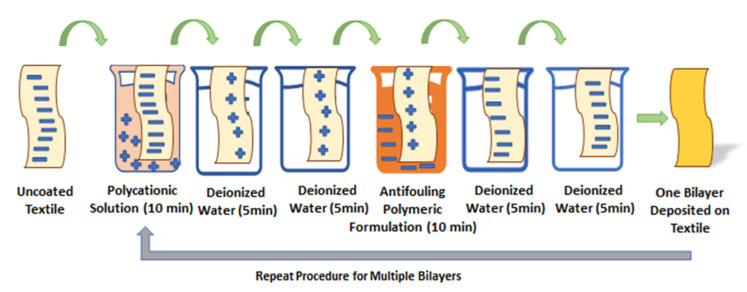
Schematic representation of fabrication of antifouling functional cotton textile through layer-by-layer technique.

**Figure 3 polymers-14-02540-f003:**
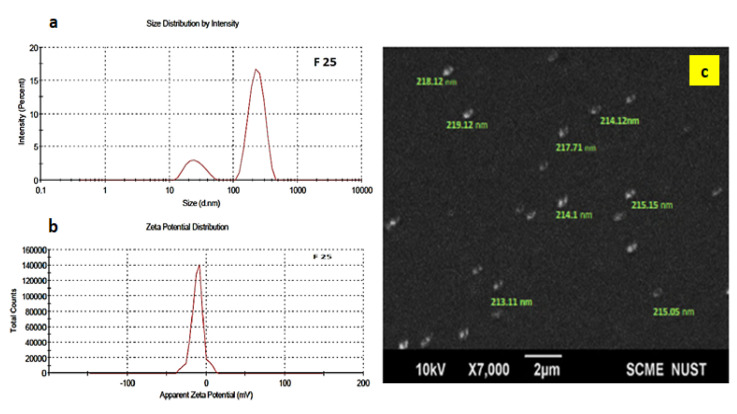
(**a**) Zeta size, (**b**) zeta potential, and (**c**) scanning electron microscope images of drug loaded nanoformulation.

**Figure 4 polymers-14-02540-f004:**
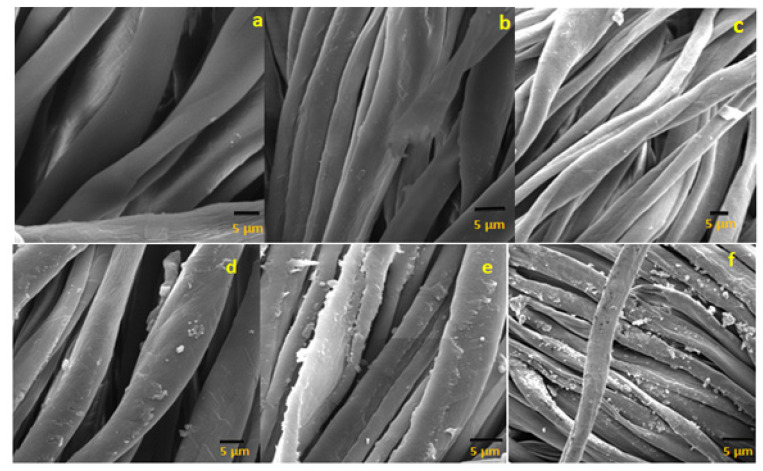
(**a**) EM images of uncoated cotton textile fabric; coated cotton textile with (**b**) 1 BL, (**c**) 5 BL, (**d**) 10 BL, (**e**) 15 BL, and (**f**) 20 BL.

**Figure 5 polymers-14-02540-f005:**
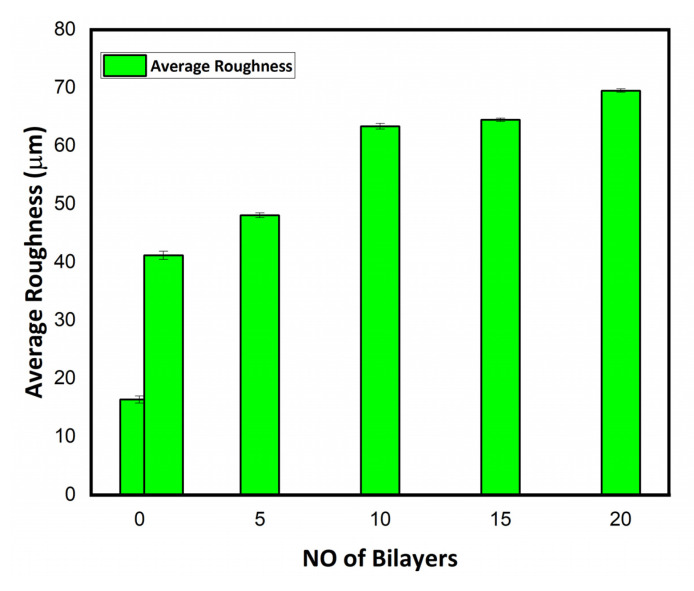
Average roughness (Ra) profile in µm of uncoated and coated cotton textile fabrics (1, 5, 10, 15, and 20 BL).

**Figure 6 polymers-14-02540-f006:**
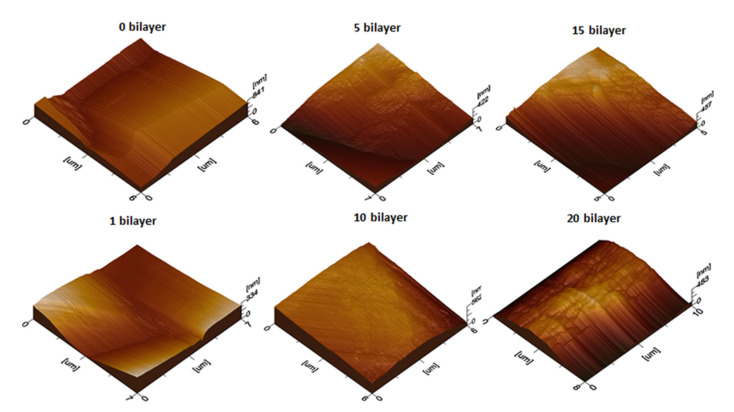
Atomic force microscopy (AFM) images of uncoated (0 BL) and coated cotton textile (1, 5, 10, 15, and 20 BL).

**Figure 7 polymers-14-02540-f007:**
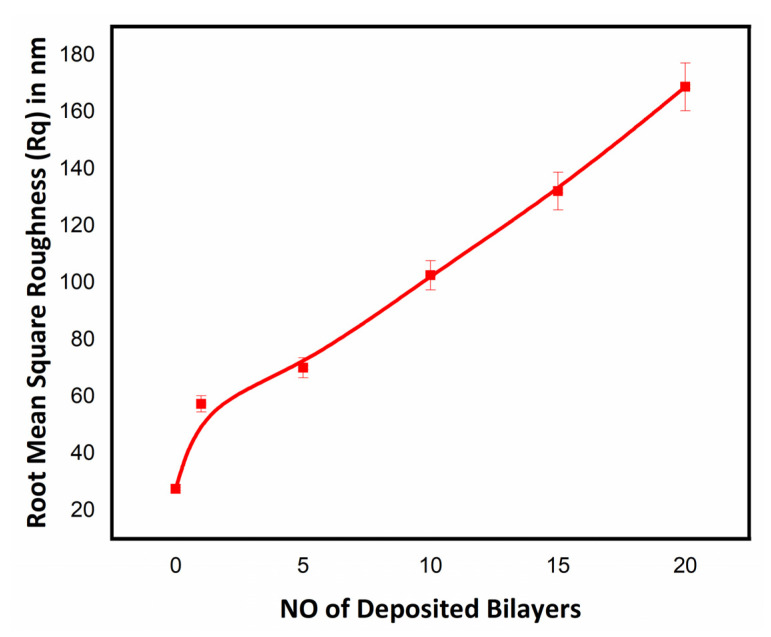
Root mean square roughness (Rq) in nm of uncoated and coated textile with deposited number of bilayers (1, 5, 10, 15, and 20 BL).

**Figure 8 polymers-14-02540-f008:**
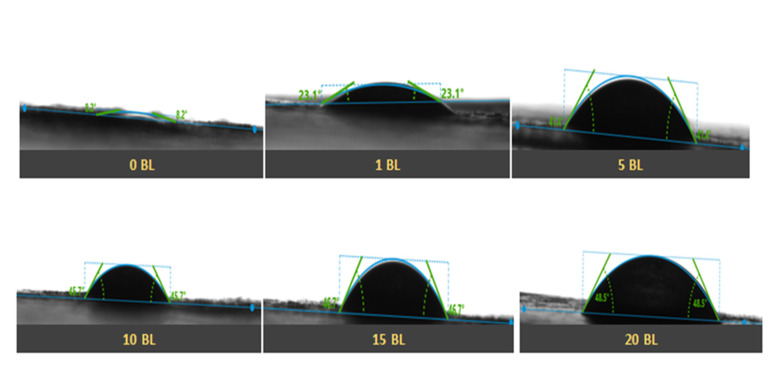
Contact angle measurement of 0, 1, 5, 10, 15, and 20 BL coated cotton textile.

**Figure 9 polymers-14-02540-f009:**
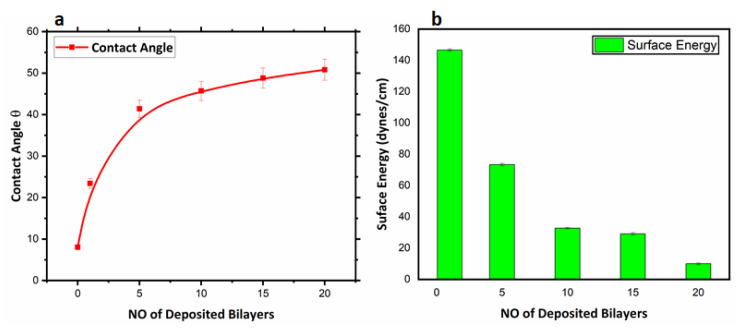
(**a**) Contact angle θ of fabricated cotton textile plot versus number of deposited bilayers; (**b**) surface energy of fabricated cotton textile plot versus number of deposited bilayers.

**Figure 10 polymers-14-02540-f010:**
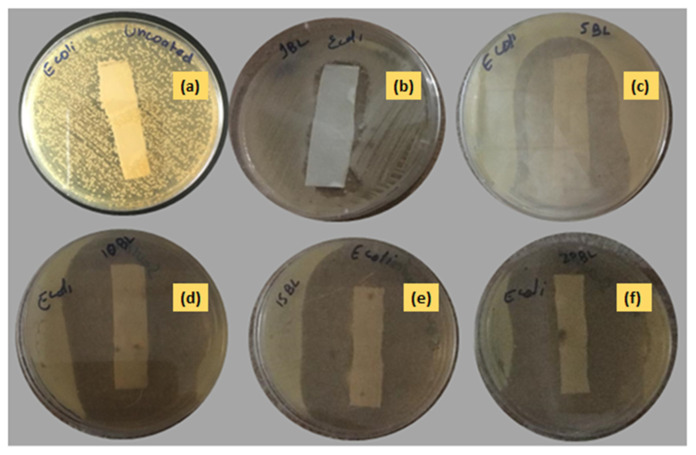
Antibacterial activity of (**a**) uncoated and coated in (**b**) 1 BL, (**c**) 5 BL, (**d**) 10 BL, (**e**) 15 BL, and (**f**) 20 BL against *Escherichia coli*.

**Figure 11 polymers-14-02540-f011:**
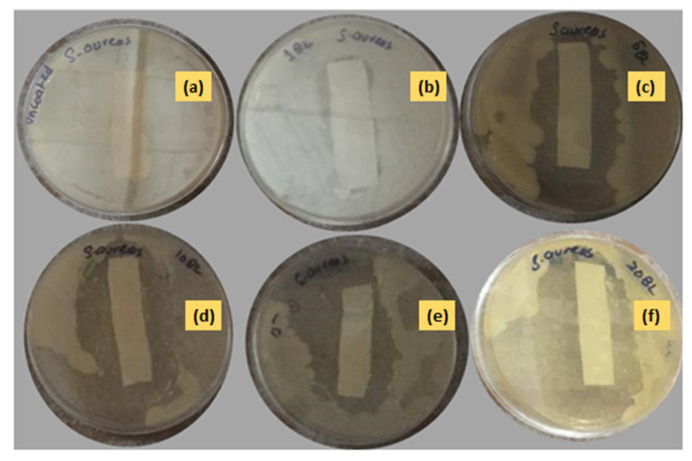
Antibacterial activity of (**a**) uncoated and coated in (**b**) 1 BL, (**c**) 5 BL, (**d**) 10 BL, (**e**) 15 BL, and (**f**) 20 BL textiles against *Staphylococcus aureus*.

**Figure 12 polymers-14-02540-f012:**
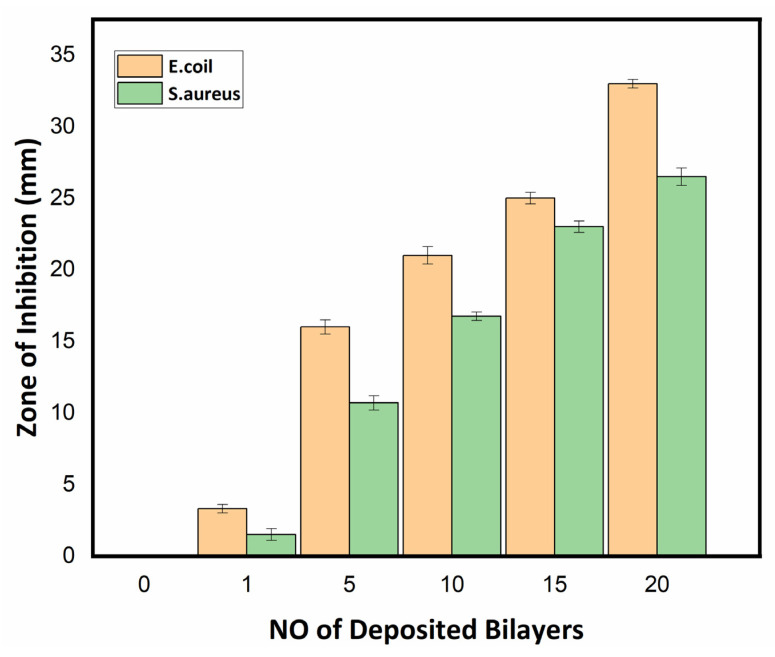
Zone of inhibition of uncoated (0 BL) and coated textiles (1, 5, 10, 15, and 20 BL) against *Escherichia coli* and *Staphylococcus aureus*.

## Data Availability

All data are available to the readers.

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
