# Peer review of "Layer-By-Layer Self-Assembled Dip Coating for Antifouling Functionalized Finishing of Cotton Textile"

_polymers, 2022, doi:10.3390/polym14132540_

Round 1

Reviewer 1 Report

The manuscript presents results on the preparation and anti-microbial/fouling textile based on a cotton fabric functionalized with multilayers of an active antifouling polymeric formulation (APF). The APF was prepared via nanoencapsulation of an antibiotic (cefotaxime) in polycaprolactone. Subsequent dip-coating of the cotton patches in a polycationic solution and APF was used to functionalize it to a multilayer structure (ranging from 1 to 20 polyDADMAC/APF bilayers). The effects of the functionalization in the as-prepared textiles were thoroughly characterised from the perspective change in their morphology, surface wettability, and anti-bacterial effects on two types of bacteria (gram-negative and gram-positive). 

The manuscript is a practical demonstration of the applicability of the authors' previous work (development and optimisation of the APF). In general, the work has positive sides in terms of novelty and practicality and it would be of interest to the audience of MDPI Polymers, and especially the Special Issue "Biomedical Applications of Polymeric Materials". However, I cannot recommend it for publication in its current state, due to its many shortcomings, as it comes to the quality of presentation and discussion/interpretation of the results, which should both be improved.

Overall, the text lacks clarity, making it inaccessible to a reader. It is bulked-up with repeating statements and lacking quantitative discussion on the results for the most part. The latter is especially notable in the Conclusions section, where virtually no quantitative summary of the effects of functionalization can be seen, but instead it seems more of summary of the Introduction. My main suggestion to the authors to carefully look through the manuscript upon re-submission and improve the clarity and quality of its presentation, from the perspective of a potential reader. 

I have also listed a few more concise comments bellow:

1) [Major] Subsection 2.1. is confusing to read and probably contains some information that can be included. Only after referring to the authors previous work [31] it becomes clear that the optimisation procedure of the composition was done there. Hence, there is no need to list the PCL masses, nor the PVA concentrations. Especially, lines 151 - 155 are very confusing /* initially I was left with the impression that after adding 3, 5, and 7 mg of cefotaxime, subsequently another 3 mg were added, before checking the previous publication */. It makes it actually unclear when and where the cefotaxime is added. Please re-write and shorten this section, focusing only on the precise composition used in the current manuscript.

2) [Major] There are many odd-wordings in the text, e.g.: “… sputtered coated with gold …” line 159; “Well-developed surface roughness in resultant nano encapsulated finished textile fabric having …” lines 265-267.

3) [Minor] In line 171 “25 mg of PCL in 10 mL optimised aqueous phase” leaves the impression that the PCL is added directly in the aqueous phase, contrary to the explanation in point 2.1. that it was added in 2 mL DCM.

4) [Question] Was dip-coating done manually? If not - may details about the withdrawal speed/rate be included in the respective section of the manuscript ?

5) [Question] The functionalization process is described in great details, including that the fabric was soaked in two separate beakers of distilled water between the polyDADMAC and the APF, however it raises some questions about the possibility of transfer of reactants between the steps, during multiple bilayer depositions. Have the authors considered and investigated this possibility ?

6) [Major] The use of two separate roughness estimates (Ra / Rq) from two separate techniques (profilometry and AFM) at different scales (um / nm) may be confusing for the reader. May the authors consider to improve the text between lines 271 - 315: firstly the increase in optical profilometry roughness by 50 um seems very odd, considering the claim that single layers of APF are added in each cycle - is it possible to add the OP images / scans from which Ra was obtained, at least for the extreme cases; secondly it is claimed that an area of 1 um (square, probably) is observed at AFM - the images in Fig. 6 show scales in 5 - 7 um instead, clarify if the Rq estimated only for a single line or a smaller part of the image (one would guess the latter, given the z-scale in these); please add a clarification in the captions of Fig. 5 and 7 about the respective technique used for each roughness measurements to help the reader; There is no unit (nm) mentioned anywhere in Fig. 7, the legend is also redundant in the case of this Fig. ( RMS Roughness (Rq) is already written on the y-axis).

7) [Minor] In Figures 10 and 11 the uncoated control is not mentioned in the figure caption. The labels, written with a sharpie, on the Petri dishes is legible, however, to make these images more professional please add digital label overlays and include them in the captions (e.g. “(a) Uncoated, (b) 1 BL, (c) 5 BL…”)

8) [Suggestion] Nothing is mentioned about the durability of the functionalized textiles, which can be considered for future work. Also - in the antibacterial assay, probably it would have been beneficial to have a control, treated with APF based only on the polymer (w/o cefotaxime addition during coprecipitation).

9) [Minor] Citation style of the references seems inconsistent with MDPI’s requirements.

Author Response

Dear professor, thank you very much for your comments and suggestions. we have modified the manuscript and made some improvements as directed. The suggested improvements and changes has highlighted in the  re-submitted file of manuscript and a cover letter attached for your consideration please. 

Reviewer 2 Report

This paper reports layer-by-layer self-assembled dip coating for antifouling functionalized finishing of cotton textile. Cotton fabric was taken as a substrate to incorporate antibacterial functionality by fabricating multilayers of antifouling polymeric formulation (APF) and polyelectrolytes solution alternatively. Fabricated textile exhibited significant zone of inhibition against both microbial strains gram positive S. aureus and 32 gram-negative E. coli. The result and discussion are interesting. However, the points in the manuscript still need some improvement. The specific comments are as follows:

  1. The size of antifouling polymeric formulation in Figure 3a should be mainly for 30-40 nm, the authors should check their result.
  2. The drug release behavior of fabricated textile should be measured.
  3. The authors should discuss the mechanism of antibacterial activity of fabricated textile in Figure 10. Which component was contributed to the antibacterial activity of fabricated textile? PDAC should also have antibacterial activity.
  4. The authors should specify the function of PVA.

Author Response

Dear professor,  thank you very much for your kinds suggestions and comments. we have made significant changes in the re-submitted file attached for your kind consideration please. It further improves our knowledge regarding present research.

Reviewer 3 Report

The manuscript of Sana Javaid et al addresses the problem of obtaining test materials with special properties. To obtain antibacterial properties, an active substance is used - cefotaxime. The paper proposes a method for creating a multilayer functional coating of cotton fabric. The authors show how the morphology of the tissue surface changes depending on the cycles of its processing. For this, the AFM and SEM methods are involved, it estimates the surface energy and the change in the contact angle. In my opinion, the manuscript has a more applied character (in my opinion, the manuscript would be better suited for the Fibers or Materials journal). The scientific novelty of the work is implicitly expressed. At the same time, the work itself was done at a good level.

Graphical abstract, Figure 5 - "No of Bilayers" should be replaced with "Number of layers"
Line 130. "TDS" - needs to be deciphered.
Line 258. "coting" needs to be corrected.
Figure 4. It is necessary to correct the quality of the drawings, the scale bar is not informative.

Author Response

Dear Professor, thank you very much for your recommendations and suggestions. we have made significant modifications and highlighted in the re-submitted file attached for your kind consideration please.

Round 2

Reviewer 1 Report

I would like to thank the author for the patience and effort in acknowledging and implementing the comments by me and the fellow referees, and apologise for any stress and inconvenience caused by any of my comments (however it is all in the best for improving the manuscript). All in all, I believe that the manuscript has been improved and is suitable for publication. I would like to wish the authors to continue their efforts on developing the topic they've selected and improving the quality of their research (and commend them for the quality of what was presented, given the circumstances). Best of luck.